# Characterization of the *FLA* Gene Family in Tomato (*Solanum lycopersicum* L.) and the Expression Analysis of *SlFLA*s in Response to Hormone and Abiotic Stresses

**DOI:** 10.3390/ijms242216063

**Published:** 2023-11-07

**Authors:** Kangding Yao, Yandong Yao, Zhiqi Ding, Xuejuan Pan, Yongqi Zheng, Yi Huang, Zhuohui Zhang, Ailing Li, Chunlei Wang, Changxia Li, Weibiao Liao

**Affiliations:** 1College of Horticulture, Gansu Agricultural University, 1 Yinmen Village, Anning District, Lanzhou 730070, China; 19119882925@163.com (K.Y.); yyd614636237@163.com (Y.Y.); 15693198831@163.com (Z.D.); panxj@st.gsau.edu.cn (X.P.); ykmzyq@163.com (Y.Z.); huangyi202309@163.com (Y.H.); 13837951879@163.com (Z.Z.); 17899315228@163.com (A.L.);; 2College of Agriculture, Guangxi University, Nanning 530004, China; licx@gxu.edu.cn

**Keywords:** fasciclin-like arabinogalactan proteins, gene expression analysis, abiotic stress, phytohormone

## Abstract

Fasciclin-like arabinogalactan proteins (FLAs), a subclass of arabinogalactan proteins (AGPs), participate in mediating plant growth, development, and response to abiotic stress. However, the characterization and function of *FLAs* in tomato are currently unknown. In this study, members of the tomato *FLA* family are characterized and analyzed in relation to their response to phytohormonal and abiotic stresses. The results show that a total of 24 *FLA* members were characterized in tomato. The structural domain analysis showed that these members have a high protein similarity. The expression profiles of different tissues indicated that the genes of most members of the tomato *FLA* gene family are highly expressed in roots, but to a lower extent in fruits. qRT-PCR analysis revealed that all 24 tomato *FLA* genes are responsive to ABA and MeJA. *SlFLAs* showed a positive response to salt and cold stress. *SlFLA1*, *SlFLA12*, and *SlFLA14* are significantly induced under darkness. *SlFLA1* and *SlFLA3* are significantly induced under drought stress. This study provides a basis for a further understanding of the role of tomato *FLA* homologous genes in plant response to abiotic stress and lays the foundation for further research on the function of *FLAs* in tomato.

## 1. Introduction

Cell-to-cell interactions and communication provide key structural, positional, and environmental signals during plant development. These signals need to pass through the plant cell wall, which surrounds the plasma membrane [1,2]. The plant cell wall is a dynamic and complex organelle that serves as the primary source of structural support and protection for plants. It is mainly composed of cellulose, hemicellulose, pectin, polysaccharides, and proteins. Additionally, the cell wall plays a significant role in signal transduction, intercellular communication, and immunity, in addition to providing mechanical protection and structural support [3,4]. Hydroxyproline-rich glycoproteins (HRGPs) are typical cell wall proteins that participate in plant growth, development, and immunity [5]. HRGPs have several repeating glycosylation motifs containing hydroxyproline (Hyp) residues as glycosylation sites. Depending on the level of glycosylation, the HRGP superfamily can be divided into three subfamilies: hyperglycosylated arabinogalactan proteins (AGPs), minimally glycosylated Pro-rich proteins (PRPs), and moderately glycosylated extensins (EXTs) [6]. AGPs are abundant in plants and can be classified into six main subclasses: classical AGPs, AG peptides, lysine-rich AGPs, fasciclin-like (FLAs), non-classical AGPs, and chimeric AGPs [7]. Fasciclin-like arabinogalactan proteins (FLAs) are characterized by fasciclin (FAS) structural domains, which were first found in fruit flies (*Drosophila melanogaster*) and later in other organisms, including algae, lichens, plants, and animals. FLAs typically have one or two bundled protein structural domains, and they are found in Drosophila, mammals, sea urchins, plants, yeast, and bacteria. In addition to the bundle protein structural domains, FLAs typically contain an N-terminal signal peptide and a C-terminal glycosylphosphatidylinositol (GPI)-anchored signal peptide. The GPI and fasciclin structural domains are functionally important and are thought to mediate cell adhesion [1,8].

To date, *FLA* family members have been identified in a wide range of plant species. A total of 21, 27, 34, 35, 19, 30, and 33 *FLA* members have been identified in Arabidopsis (*Arabidopsis thaliana*, At) [1], rice (*Oryza sativa*, Os) [9], wheat (*Triticum aestivum*, Ta) [2], poplar (*Populus trichocarpa*, Ptr) [10], cotton (*Gossypium hirsutum*, Gh) [11], pepper (*Capsicum annuum*) [12], and cabbage (*Brassica rapa*, Br) [13], respectively. *FLAs* are cell wall structural glycoproteins that mediate cellulose deposition and cell wall development. They are thought to be involved in fiber development, elongation, and stem dynamics. *FLAs* affect fiber and wood quality in cotton and woody plants, such as poplar and eucalyptus (*Eucalyptus globules* Labill, Egr) [14], and they are abundant in xylem [15]. Studies on plant *FLAs* have focused on tissue-specific functions, pollination, and embryogenesis, and on general responses to biotic and abiotic stresses [10,16]. In addition, many studies have demonstrated the importance of *FLAs* in cell wall biosynthesis. For example, in *A*. *thaliana*, *AtFLA4* induced aberrant cell expansion, adhesion, cell wall synthesis, and seed coat pectin mucilage [17]. *AtFLA11* and *AtFLA12* play a role in stem tensile strength, biomechanics, and elastic modulus, thereby affecting cell wall composition and structure [18]. Studies on *AtFLA16* mutants have shown that the deletion of *AtFLA16* resulted in a shorter stem length and altered biomechanical properties [19]. The *PtFLA6* gene was associated with xylem fiber cells, cell wall composition, and stem biomechanics in poplar [14,20]. Stem lignification is more developed in herbaceous plants, and lignified stems are mainly composed of secondary wall-enhancing cells. *FLAs* are cell wall structural glycoproteins that mediate cellulose deposition and cell wall development. They are believed to participate in fiber development, elongation, and stem dynamics [14].

Tomato (*Solanum lycopersicum* L.) has long been used as a model plant for fruit ripening, disease response, genetics, and whole-genome sequence studies [21]. Abiotic stresses can adversely affect yield, productivity, and quality in tomato [22]. To date, *FLAs* have been studied in many plants, but reports of *FLAs* in tomato are relatively rare. Therefore, to better understand the key role of the *FLA* family in plants, the coding genes of tomato *FLA* subfamily members are characterized and analyzed in this study. The tomato *FLA* members were analyzed for gene structure, secondary structure, chromosomal location, conserved motif analysis, cis-acting element analysis, phylogenetic tree, and subcellular localization. The expression patterns of these *FLAs* in various tissue-specific processes and gene transcription analysis under different abiotic stresses and hormonal conditions are also investigated. The objective of this study is to lay the foundation for future research on the role of *FLAs* in tomato growth, development, and resistance to stresses.

## 2. Results

### 2.1. Identification of FLA Genes in Tomato

A total of 25 protein sequences containing the *FLA* structural domain (E-value = 1 × 10^−5^) were retrieved from the Tomato Genome Database. The incomplete proteins (not containing start or stop codons) and proteins whose annotation information in Blastp (database SSWISS-PROT/Uniprot (accessed on 5 June 2023)) in NCBI did not contain *FLA* structural domains were screened out, and 24 FLA proteins were finally characterized (Table 1). They were sequentially named *SlFLA1*~*SlFLA24*. Using tomato genome annotation information and the TBtools software, we visualized the chromosomal distributions of the tomato *FLA* gene family members. As shown in Figure 1, tomato *FLA* genes are unevenly distributed on 10 chromosomes, where the number of genes on each chromosome is independent of the chromosome size. The average amino acid number of the tomato *FLA* family is 339, the molecular weight is in the range of 25.79 kDa~49.43 kDa, and the isoelectric point (pI) is lower than 7, except for *SlFLA9*, *SlFLA17*, *SlFLA18*, and *SlFLA24*, in which it is greater than 7. The average value of the instability index is 44.33, and the mean value of the aliphatic index is 91.42. The maximum value of the grand average of hydropathicity is 0.289 and the minimum value is −0.29; within this range, the members with a grand average of hydropathicity of less than 0 account for 50%. The results of the subcellular localization prediction indicate that chloroplasts contain the most members of the tomato *FLA* gene family, with nine members localized to chloroplasts, followed by vesicles and extracellular material (Table 1). The signal peptide prediction results indicate that all members of the tomato *FLA* family, except *SlFLA19*, have signal peptides with loci located around 20. The results of the transmembrane structure prediction indicate that all members of the tomato *FLA* family have transmembrane structures (Table 1).

### 2.2. Structural Analysis and Phylogenetic Tree Analysis of the SlFLA Gene Family

Our analysis revealed that 25% of the tomato *FLA* family members, namely, *SlFLA1*, *SlFLA5*, *SlFLA18*, *SlFLA19*, *SlFLA21*, and *SlFLA24*, contain one intron; it is worth noting that *SlFLA1* and *SlFLA5* have longer introns, and the remaining family members do not have any introns (Figure 2). To investigate the selection pressure during the evolution of *SlFLA* genes, the Ka/Ks values of tomato *FLAs* were calculated (Appendix A). The results show that the Ka/Ks ratio was lower than 1 was 59% for *SlFLA* genes, indicating that the *SlFLA* gene family might be selected for purification.

FLA proteins from these plants were divided into four subgroups based on homology (A, B, C, and D; Figure 3), in which group A contained the smallest number of members (5 members, with 1 tomato *FLA* member); group B (22 members) contained 7 *FLA* members, one branch of which was the pepper *FLA* family in addition to *SlFLA5* and a further 8 members; and group C (13 members) contained only tomato and Arabidopsis *FLA* members, of which the closest relatives were Arabidopsis *AT1G03870.1* and Arabidopsis *AT5G44130.1*. Group D (35 members) contained the largest number of members, and the closest relatives were tomato *SlFLA4* and *SlFLA5*, *SlFLA13*, and *SlFLA3*. In addition, among the four subgroups, the plant most closely related to tomato was pepper; so, we hypothesized that tomato and pepper have a certain evolutionary relationship. Meanwhile, members of the tomato *FLA* family are closely related to Arabidopsis and pepper (Figure 3).

### 2.3. Analysis of Cis-Acting Elements of Tomato FLA Family Genes

The tomato *FLA* gene contains a total of 25 homeotic-acting elements (Figure 4 and Table 2). Among them, eleven elements (AE-box, ATCT-motif, Box 4, MRE, GA-motif, TCT-motif, TCCC-motif, GATA-motif, G-box, GT1-motif, and LAMP-element) are associated with light response, four elements (LTR, MBS, ARE, and TC-rich repeats) are related to stress response, and nine elements (P-box, TATC-box, GARE-motif, and TCA-element) are related to hormone response (Figure 4). To further investigate the cis-elements in the *FLA* promoter sequence, three major cis-acting elements were identified, including light-, stress-, and hormone-acting elements (Figure 5). Box 4 element is distributed in all tomato *FLA* genes, except *SlFLA3*, and *SlFLA22* contains the highest number. The GA-motif is the least light-responsive element. The LTR element is mainly distributed in *SlFLA1*, *SlFLA12*, *SlFLA4*, and *SlFLA18*. The ARE element is mainly distributed in *SlFLA1* and *SlFLA6*. Both CGTCA-motif and TGACG-motif elements are mainly distributed in *SlFLA15*. Thus, the relative abundance of cis-elements related to light, stress, and hormones suggests that the tomato *FLA* gene might play a crucial role in regulating plant growth and hormone responses.

### 2.4. Conserved Domain and Conserved Motifs of the Tomato FLA family

The conserved structural domains of the proteins encoded by tomato *FLA* was analyzed. The results show that the FLA protein has a conserved protein structure including three conserved domains of H1, H2, and [Y/F] H at the N-terminus (Figure 6A) [11]. The tomato FLA protein has a conserved fasciclin structural domain. In the tomato FLA family, the maximum value was set to 10 conserved motifs (Figure 6B), and the sequence information of the characterized conserved motifs is shown in Table 3, where the amino acid sequences of the different conserved motifs are indicated by a stack of letters at each position (Figure 6C). The length of each motif varies between 10 and 50 amino acids. The results show that the 10 characterized tomato FLA motifs are very similar, with Motif4 presented in all tomato FLA proteins. In addition, Motif4, Motif7, and Motif8 occur twice in some members.

### 2.5. Analysis of the Protein Secondary Structure of Tomato FLA Family Genes

The most abundant protein secondary structures in tomato FLA members were mainly α-helices and random coils (Table 4). The 24 FLA-encoded protein secondary structures were α-helices (22.6–41.43%), extended chains (13.15–25.99%), β-turns (2.41–8.01%), and random coils (36.69–53.41%).

### 2.6. Tissue-Specific Expression Pattern of Tomato FLA Genes

In order to study the expression of *FLA* genes in tomato tissues at different growth stages, we analyzed the expression of *FLA* genes in 16 tomato tissues, including unopened flower buds, fully opened flowers, leaves, roots, 1 cm fruits, 2 cm fruits, 3 cm fruits, ripened green fruits, pink fruits, red fruits, breaker fruit, breaker fruit+5, breaker fruit+7, breaker fruit+10, and ripened fruits (Figure 7). *SlFLA* expression was relatively high in root, stem, leaf, flower, and green fruit, whereas the expression was lower in fruits at the breaker fruit and ripening stages. Most *SlFLAs* were most highly expressed in roots, whereas *SlFLA8*, *SlFLA9*, *SlFLA19,* and *SlFLA20* were most highly expressed in flower buds. In addition, *SlFLA1* expression was significantly higher in fruits than other genes. But *SlFLA14* had a low expression level in all tissues. Thus, the *SlFLA* family is mainly expressed in roots, followed by leaves and stems.

### 2.7. Expression of Tomato FLA Genes in Response to MeJA and ABA Treatments

In order to clarify the role of *FLAs* in tomato under hormone stress, the expression of 12 tomato *FLA* genes under different treatments was determined based on the statistical analysis of cis-acting elements. Its most hormone-responsive elements are MeJA (methyl jasmonate) and ABA (abscisic acid) based on the results of cis-acting element analysis, and thus MeJA and ABA were used to treat tomato seedlings. The tomato *FLA* family responded to MeJA and ABA to varying degrees (Figure 8). *SlFLA1* was significantly up-regulated after 6 h of MeJA and ABA treatments. Under the MeJA treatment, the expression of all members, except *SlFLA5* and *SlFLA7*, was significantly up-regulated at 6 h. In contrast, *SlFLA1* reached the highest levels at 12 h of treatment with MeJA. Under the ABA treatment, the expression of all members, except *SlFLA7*, was significantly up-regulated at 6 h; *SlFLA7* increased by 6.5-fold at 6 h compared with at 0 h. *SlFLA1* reached its highest value at 12 h, and both *SlFLA3* and *SlFLA7* increased gradually, reaching their highest levels at 24 h. Interestingly, the expression of *SlFLA14* and *SlFLA21* under the ABA treatment showed a gradual increase over time. Therefore, most of the tomato *FLA* genes were significantly up-regulated under the stimulation of ABA and MeJA, and their expression reached the highest value at 6 h, and then gradually decreased to the lowest at 48 h.

### 2.8. Expression Profile Analysis of FLA Genes in Tomato under NaCl, Dark, Cold, and PEG Treatments

To elucidate the role of *FLAs* in tomato under abiotic stress, the expression levels of 12 *FLA* genes were investigated in tomato under low temperature, darkness, NaCl, and PEG (Polyethylene glycol) treatments. As shown in Figure 9, the relative expression of the 12 *SlFLA* genes was different under low-temperature conditions. Under low-temperature conditions, the expression of all members, except *SlFLA15*, was significantly up-regulated at 6 h (Figure 9A); *SlFLA15* was down-regulated by 0.3-fold at 6 h compared with at 0 h. Among all members, *SlFLA3* reached its highest value at 24 h; the expression levels of the remaining genes were highest at 24 h and then declined slowly. The greatest changes were observed in *SlFLA1* and *SlFLA14* in the low-temperature treatment. In addition, *SlFLA1* and *SlFLA14* were expressed at higher levels than other members under darkness, low temperature, and NaCl stresses; their expression increased by at least 500-fold at 6 h compared with at 0 h.

Under darkness stress, *SlFLAs* were significantly upregulated at 6 h, except for *SlFLA11* (Figure 9B); *SlFLA21* reached its highest value at 24 h and other members reached their highest values at 6 h. Tomato *FLA* expression was up-regulated by NaCl treatment (Figure 9C). The expression levels of all members, except *SlFLA1*, reached their highest values at 6 h. The relative expression of *SlFLA12* and *SlFLA14* remained unchanged after 12 h, whereas the expression of the remaining *SlFLA* genes gradually decreased. The greatest change in expression was observed in *SlFLA14*, which increased by 470.3-fold at 6 h compared with at 0 h. In addition, *SlFLA* was significantly up-regulated to a higher extent under NaCl stress compared with other abiotic stresses (dark, PEG, and low temperature). Thus, we predict that tomato *FLA* mainly responds to NaCl stress.

The relative expression of the twelve tomato *FLA* genes showed a similar trend under PEG treatment (Figure 9D). Ten of these genes (*SlFLA1*, *SlFLA3*, *SlFLA5*, *SlFLA11*, *SlFLA21*, *SlFLA14*, *SlFLA15*, *SlFLA17*, *SlFLA18*, and *SlFLA20*) were up-regulated under short-term (6 h) PEG stress. The greatest change in expression was observed in *SlFLA1*, *SlFLA3*, and *SlFLA12*, which increased by 522.11-fold, 55.96-fold, and 154.81-fold, respectively, compared with that at 0 h. With the increase in drought treatment time, the expression of *SlFLA1* and *SlFLA14* tended towards stability.

## 3. Discussion

FLAs are a subdivided type of HRGPs, which are typical cell wall proteins involved in plant growth and development and immunity [5]. *FLAs* may be involved in plant growth and development and adaptation to environmental conditions [10]. In total, 21 *FLA* genes have been identified in *A*. *thaliana* [1], 27 *FLA* genes have been identified in rice [9], 34 *FLA* genes have been identified in wheat [2], 35 *FLA* genes have been identified in poplar [10], and 19 *FLA* genes have been identified in cotton [11]. In this study, a more comprehensive and systematic analysis of the *FLA* gene family in tomato was carried out using bioinformatic techniques, and 24 *FLA* genes were characterized and shown to be distributed on 10 chromosomes in tomato (Figure 1).

In addition, we found that their structural domains were conserved (Figure 6). Of the 24 *SlFLAs*, 17 had a single-bundle protein structural domain and 7 had two structural domains. The signal peptide prediction indicated that all members of the tomato *FLA* family, except for *SlFLA19*, had signal peptides, and loci located around 20. Additionally, 24 tomato *FLAs* are GPI-anchored. Most putative *GhFLAs* have a C-terminal signal sequence recognized by a transamidase that replaces this peptide sequence with a GPI module [11]. GPI-anchored signals with bundle protein structural domains are important for cell adhesion, membrane localization, and enabling more stable interactions between adhesion complexes. However, there is no consistent pattern in the number of fasciclin domains or the presence of a GPI signal. It has been suggested that plants may have *FLAs* that are GPI-anchored to maintain the integrity of the plasma membrane, whereas *FLAs* that are not GPI-anchored are used to mediate cell expansion [5].

We analyzed the structure of 24 *SlFLA* genes and found that 25% of *SlFLAs* contain one intron, whereas the remaining members have lost their introns (Figure 3). In *Salicornia spp*., 34.88% of *FLA* genes contain one intron [23], and a similar structure occurs in tobacco, where the majority of the members do not have introns [24]. The introns of *FLA* genes are normally lost during plant evolution [23]. The previously reported fasciclin structural domain contains about 110–150 amino acid residues and has two highly conserved regions (H1 and H2) and a [Phe/Tyr]-His ([Y/F] H) motif [11], which has been found in tobacco, wheat, Arabidopsis, and rice [2,24]. We found that tomato *FLAs* also possess this structural domain (Figure 6A), which is similar to previous identifications in other species, suggesting that the members of the tomato *FLA* gene family are conserved in gene structure. As in the case of the *FLAs* found in other species, these 24 *SlFLAs* contain the conserved structural domains of typical FLA proteins. Therefore, fasciclin domains are important for the function of the molecule.

To understand the relationship between tomato FLA members and other species, we constructed a phylogenetic tree of *FLA* members in tomato, Arabidopsis, and pepper, and we classified 75 FLA members into four groups. The 24 *SlFLA* members belonged to these four types (Figure 3), which indicates that the structure and function of *SlFLAs* are highly conserved during plant evolution [25]. In group D, the number of *FLA* members in tomato and other species was greater than the number of members distributed in groups A, B, and C, suggesting that *FLAs* underwent rapid adaptive evolution in group D [26]. In addition, in the same group, the closest members have similar gene structures and may have similar functions. It has been demonstrated that *AtFLA1* and *AtFLA3* play important roles in Arabidopsis in response to low temperature and salt stress [27]. Therefore, we speculated that *SlFLA1* might have a similar function to these genes. Subsequent experiments also demonstrated that most of the *SlFLA* genes were indeed up-regulated under salt stress (Figure 9). Related studies have found a correlation between the abundance of *AtFLA11* and *AtFLA12* transcripts containing a single FAS domain and the onset of secondary cell wall cellulose synthase expression in Arabidopsis stems [18]. In addition, the phenotypes of *AtFLA11* mutants showed the presence of a mild collapsed vessel phenotype and reduced stem cellulose content [18]. These analyses indicated that the *FLA* members in group C are associated with secondary wall and cellulose synthesis in the stem.

Previous studies have shown that *FLAs* are expressed in different patterns in various plant tissues [18,24,28]. For example, *AtFLA11/12* and some *EgrFLAs* were highly expressed in stems [18], 10 *Pop-FLAs* were highly expressed in poplar stems [28], *NbFLA11*/*18*/*31*/*32*/*34* were highly expressed in young leaves, and *NbFLA4* was highly expressed in flowers [24]. In addition, some *FLAs* were highly expressed in roots, for example, *PtrFLA12/21/22/24/27/28/30* and *NbFLA7/34* [10,24]. In the present study, we found that most of the members of the tomato *FLA* gene family were highly expressed in roots (Figure 7), including *SlFLA6/7/16/18/21/24*, suggesting that *SlFLAs* might be involved in the development of root apical meristematic tissue. Previously, it was reported that *PtFLA6* was specifically expressed in tension wood, and the reduction in *PtFLA6* transcripts affected trunk dynamics [20]. It was suggested that Arabidopsis *FLA11* and *FLA12* might have an effect on plant stem strength and stem elastic modulus [18]. Indeed, our results show that *SlFLA11* (homologue *AtFLA11/12*) was highly expressed in stems (Figure 7), and it was hypothesized that *SlFLA11* may have the same function as *AtFLA11/12* in Arabidopsis, indicating that it may play a role in stem dynamics. For tomato, lodging resistance is essential in order to avoid yield losses, and has become one of the main goals of crop breeding [29]. Considering that Arabidopsis *AtFLA11* and *AtFLA12* are involved in plant stem strength and stem elasticity modulus, analyzing the homologues of these genes (*SlFLA11*) may help to reveal their roles in regulating tomato stem strength and ultimately contribute to the breeding of tomato varieties.

Several biotic and abiotic stresses result in significant changes in the transcription of *FLAs*. For example, the expression level of wheat *FLA* proteins was elevated under H_2_O_2_ stress, which may contribute to H_2_O_2_ tolerance in wheat [30]. Similarly, the expression of *AtFLA3* was enhanced by cold stress [31]. In this study, we found that all *SlFLA* expression levels were elevated at 6 h and relatively reduced at 12 to 24 h after exposure to low temperature or salt stress. *PtrFLA2/12/20/21/24/30* expression was up-regulated under salt stress [10], and *OsFLA10/18* expression was reduced [9]. Interestingly, cotton *FLAs* were down-regulated under salt stress [11], whereas our results show that tomato *FLAs* were all up-regulated after 6 h of salt stress. We hypothesize that this could be due to an initial stress response triggered by low temperature or salt stress, followed by a subsequent adaptation phase. However, further studies are needed to confirm this.

In addition, *TaFLA3/4/9* were down-regulated after either ABA or NaCl treatment [2], and *OsFLA24* and *AtFLA1/2/8* were significantly reduced by the exogenous ABA treatment [8,9]. In this paper, we found that *SlFLA15* and *SlFLA17* were down-regulated by the ABA treatment (Figure 8A), suggesting that they are involved in ABA signal transduction pathways. In addition, this study showed that tomato *FLAs* were also up-regulated by the MeJA treatment, with *SlFLA1* being the most significant, and that both drought and darkness caused different degrees of up-regulation of tomato *FLAs*. *TaFLA9/12/14* were specifically upregulated by dehydration stress [2]. Therefore, the tomato *FLA* gene family has important potential functions for growth and abiotic stress response, whereas the specific functions of tomato *FLA* genes need to be investigated in depth.

## 4. Materials and Methods

### 4.1. Identification of the FLA Family Members in Tomato

The whole-genome data (SL3.0) and annotation files (SL3.0) for tomato were downloaded from NCBI (https://www.NCBI.nlm.nih.gov/ (accessed on 5 June 2023)). Tomato CDS sequences were extracted through the “Gtf/Gff3 Sequences Extract” feature of TBtools software (v1.09876), and the software’s “Batch TranSlate CDS to Protein” function was then used to convert CDSs to protein sequences [28]. The Arabidopsis database was used to search for the identified *FLA* gene family members, and the ID and protein sequences of the Arabidopsis *FLA* gene family were saved. To determine further whether an identified protein belonged to the *FLA* gene family, the “Batch Web CD-Search Tool” function of NCBI and the “Visualize NCBI CDD Domain Pattern” function of TBtools were used to analyze the protein domain, and those that did not contain *FLA* domains were deleted. Finally, the members of the cucumber *FLA* gene family were obtained, and these genes were named *SlFLA* genes.

### 4.2. Physicochemical Properties and Signal Peptide Analysis of the FLA Gene Family in Tomato

To analyze the physicochemical properties of the tomato *FLA* gene family, the molecular weights, instability coefficients, isoelectric points, and hydrophilicity of each member of the horse tomato *FLA* gene were analyzed using the Expasy online website (https://web.expasy.org/compute_pi/ (accessed on 8 June 2023)). The WoLF PSORT online website (https://wolfpsort.hgc.jp/ (accessed on 8 June 2023)) was used to analyze the subcellular localization of each *SlFLA*. The signal peptide prediction of the tomato FLA protein was performed using SignalP (https://services.healthtech.dtu.dk/service.php?SignalP-5.0 (accessed on 8 June 2023)).

### 4.3. Gene Location, Ka (Nonsynonymous)/Ks (Synonymous) Analysis, and Gene Structure Analysis

The chromosomal position distribution of tomato *FLA* genes was analyzed using the “Gene Location Visualize from GTF/GFF” function of TBtools software [32]. The “Simple Ka/Ks Calculator (NG)” function of TBtools software was used to calculate the selection and evolutionary pressure values of the tomato *FLA* gene family.

### 4.4. Conserved Motif and Protein Conserved Domain Analysis

The shared conserved motifs of the tomato *FLA* gene family were analyzed online using the MEME website (https://meme-suite.org/meme/tools/meme (accessed on 10 June 2023)), and the results were visualized using the TBtools software; the structural visualization of the tomato *FLA* gene family was undertaken using the TBtools “Gene Structure View (Advanced)”. The conserved structural domains of the tomato *FLA* gene family were analyzed using the DNAMAN software (v6).

### 4.5. Phylogenetic Tree and Cis-Acting Elements Analysis

The FLA protein sequences of Arabidopsis and pepper (*Capsicum annuum*) were downloaded from the Arabidopsis and NCBI databases, respectively. The protein sequences of tomato, Arabidopsis, and pepper were combined in the same file, and the evolutionary tree was constructed using the MEGA11 software (v11.0.13), in which the neighbor-joining method was used; the number of replicates was set to 1000, and the rest of the options were set to the default values. The website Evolview (https://www.evolgenius.info//evolview/#mytrees/clcle/123 (accessed on 12 June 2023)) was then used for the further modification of the evolutionary tree.

The “Gtf/Gff3 Sequences Extract” and “Fasta Extract (Recommended)” functions of TBtools were used to extract the 2000 upstream *FLA* genes of tomato from the tomato gene databases. Here, 2000 bp of data upstream of the tomato *FLA* genes were extracted from the tomato gene databases; these were submitted to the PlantCARE (http://bioinformatics.psb.ugent.be/webtools/plantcare/ (accessed on 26 June 2023)) database for gene homeotic element (promoter) analysis and were visualized using TBtools.

### 4.6. Tissue Expression Analysis of the FLA Genes in Tomato

The IDs of the *SlFLA* genes were searched in the TomExpress (http://tomexpress.toulouse.inra.fr/ (accessed on 4 July 2023)) database. Then, the data were sorted and the expression patterns of *SlFLA* in different tissues were drawn using TBtools [33].

### 4.7. Transcriptional Analysis of the FLA Genes in Tomato under Different Abiotic Stresses and Hormone Treatments, Plant Materials, and Treatments

Tomato (*S. lycopersicum* ’Micro-Tom’) seeds were provided by Gansu Agricultural University. The sterilized seeds were put into a 250 mL conical flask filled with 100 mL sterile water, then placed in a HYG-C type shaker, and cultured at a rotation speed of 180 r min^−1^. The sterile water was changed once a day. After germination, tomato seeds were transferred to plug trays containing substrate. After the cotyledons were fully expanded, a nutrient solution was added to the plant container every 2 d. The environmental photoperiod of the control growth chamber was 16/8 h (light/dark), the air temperature was 26/20 °C (day/night), and the light intensity was 250 µmol m^−2^ s^−1^. After 21 d, healthy seedlings of uniform size were selected for subsequent treatments. For the NaCl, ABA, MeJA, and PEG 6000 treatments, the seedlings were transplanted into 1/2 Hoagland nutrient solution containing NaCl (150 mM), ABA (100 µM), MeJA (50 µM), and PEG 6000 (20%) and incubated for 0, 6, 12, 24, and 48 h. For the cold treatments, the seedlings were placed in 1/2 Hoagland nutrient solution and in a refrigerator (Qingdao Haier Specialty Appliances Co., Ltd., Qingdao, China) at 4 °C for 0, 6, 12, 24, and 48 h. For the dark treatment, the tomato seedlings were treated in black airtight and breathable paper boxes. Leaf samples were collected for qRT-PCR experiments after treatment at 0, 6, 12, 24, and 48 h [34].

The collected samples were immediately frozen with liquid nitrogen and stored in a vertical ultra-low-temperature refrigerator at −80 °C (Qingdao Haier Special Electric Appliance Co., Ltd., Qingdao, China). Each treatment contained three biological replicates.

### 4.8. RNA Extraction and qRT-PCR Fluorescence Quantification

Total RNA was extracted from the samples using TRIzol reagent (Invitrogen, Carlsbad, CA, USA), taking advantage of the FastQuant First Strand cDNA Synthesis Kit (Tiainen, Beijing, China) to synthesize cDNA. These reactions were carried out under the following conditions: 37 °C for 15 min, 85 °C for 5 s, and finally ending at 4 °C. LightCycler 480 Real-Time PCR System (Roche Applied Science, Penzberg, Germany) and SYBR Green Premix Pro Taq HS Premix kit were used for qRT-PCR. The reaction system was 2×SYBR Green Pro Taq HS Premix 10 mL, primer F 0.4 μL, primer R 0.4 μL, cDNA 2 μL, and ddH_2_O 7.2 μL. The primers used in qRT-PCR were designed with Primer 5.0, and the internal reference was Actin (NC015447.3) as shown in Appendix A. The qRT-PCR data were analyzed using the 2^−ΔΔCt^ calculation method.

### 4.9. Data Statistics and Analysis

The data were analyzed using the SPSS 22.0 software (SPPS Inc., Chicago, IL, USA). In all experiments, at least three biological replicates were used. All data from three independent experiments were used. Statistical differences between measurements at different times or in different treatments were analyzed using Duncan’s multiple range test. Differences were considered significant at a probability level of *p* < 0.05 [35].

## 5. Conclusions

In general, 24 *FLA* genes were characterized in the tomato genome. This gene family contains a common conserved structural domain, and its members all contain signal peptides and transmembrane structures. Furthermore, *SlFLA* members might have key housekeeping functions by regulating cellular metabolism during plant growth and development. Last but not least, we provide evidence that tomato *FLAs* might be involved in mitigating various abiotic stresses and hormonal responses, and we predict that tomato *FLAs* mainly respond to NaCl stress, with *SlFLA1* showing the greatest changes in expression under hormonal and abiotic stresses. *SlFLA1* and *SlFLA3* are significantly induced under drought stress. Thus, the present study may provide support for further studies on the involvement of the *SlFLA* gene family in tomato growth regulation and stress response, and provide a theoretical basis for the further exploration of the functions of plant *FLA* members.

## Figures and Tables

**Figure 1 ijms-24-16063-f001:**
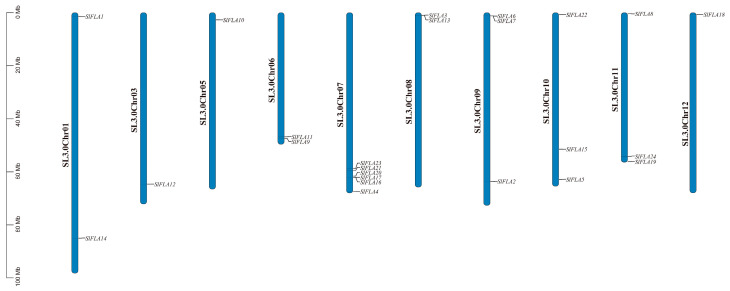
Chromosomal localization of the *FLA* gene in tomato. Chromosome positioning was based on the physical location of the 24 tomato *FLA*s. Chromosome numbers are shown at the top of each bar chart. Gene names are indicated in black. The scale bar is on the left.

**Figure 2 ijms-24-16063-f002:**
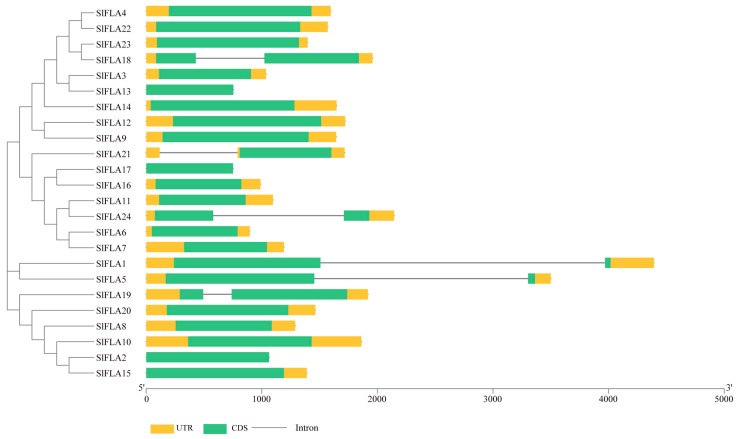
Exon-intron structure of the *FLA* gene family in tomato. The evolutionary tree was constructed based on the full lengths of tomato *FLA* protein sequences using MEGA11.0 (v11.0.13). The exon-intron graph of tomato *FLA* genes was drawn using TBtools software (v1.09876). The untranslated regions (UTRs) are indicated by thick yellow boxes; the exons are indicated by thick green boxes; the introns are indicated by black lines.

**Figure 3 ijms-24-16063-f003:**
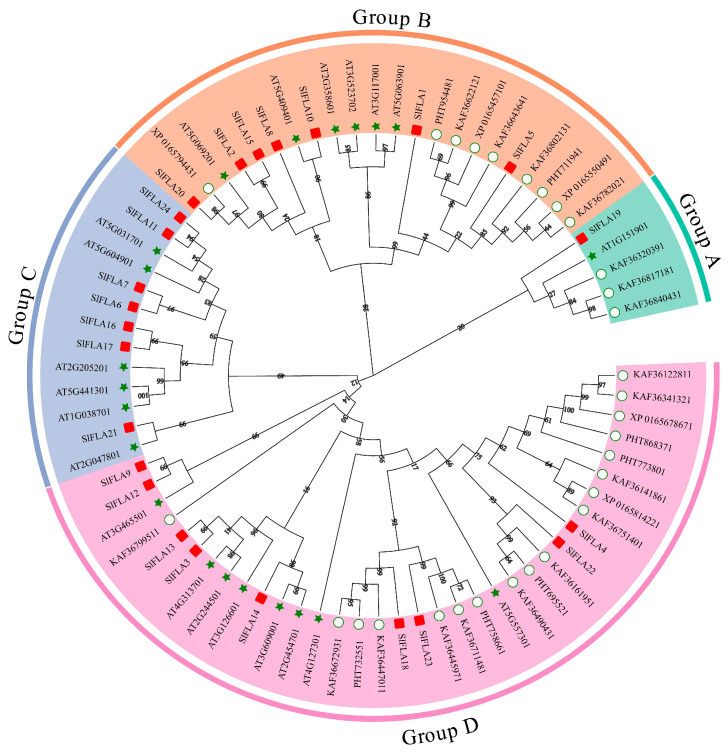
The unrooted phylogenetic tree of the *FLA* gene family in *S. lycopersicum*, Arabidopsis, and *Capsicum annuum*. The maximum likelihood method was used to construct a phylogenetic tree containing 24 tomato, 21 *A*. *thaliana* (At), and 30 pepper FLA proteins. The four subgroups are colored differently. The three differently colored shapes represent FLA proteins from three species. The red rectangles are *S. lycopersicum*, the green stars are Arabidopsis, and the white circles are *Capsicum annuum*. The number on the node in the phylogenetic tree represents the percentage of trustworthiness of the branch in the bootstrap validation.

**Figure 4 ijms-24-16063-f004:**
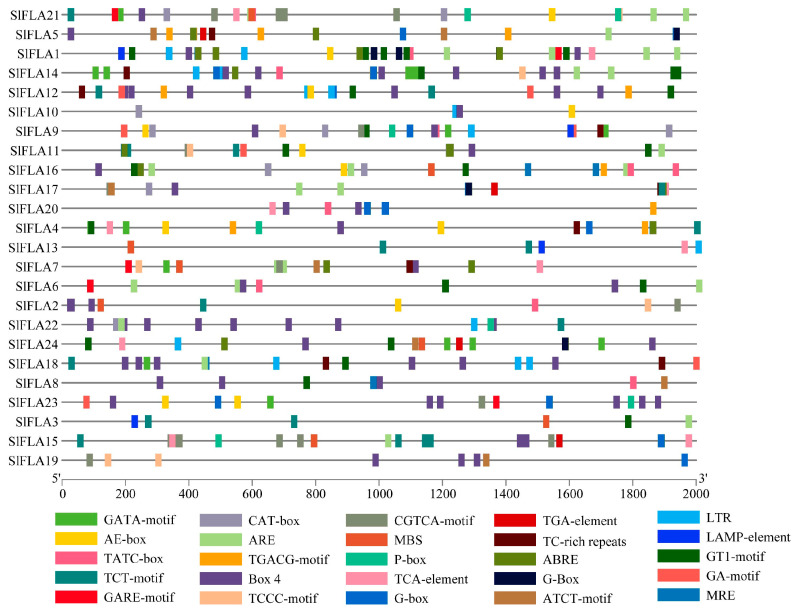
Analysis of cis-acting elements of the FLA gene family in tomato. Different colored wedges represent different cis elements. The length and position of each *SlFLA* gene were mapped to scale. The scale bar represents the length of the DNA sequence.

**Figure 5 ijms-24-16063-f005:**
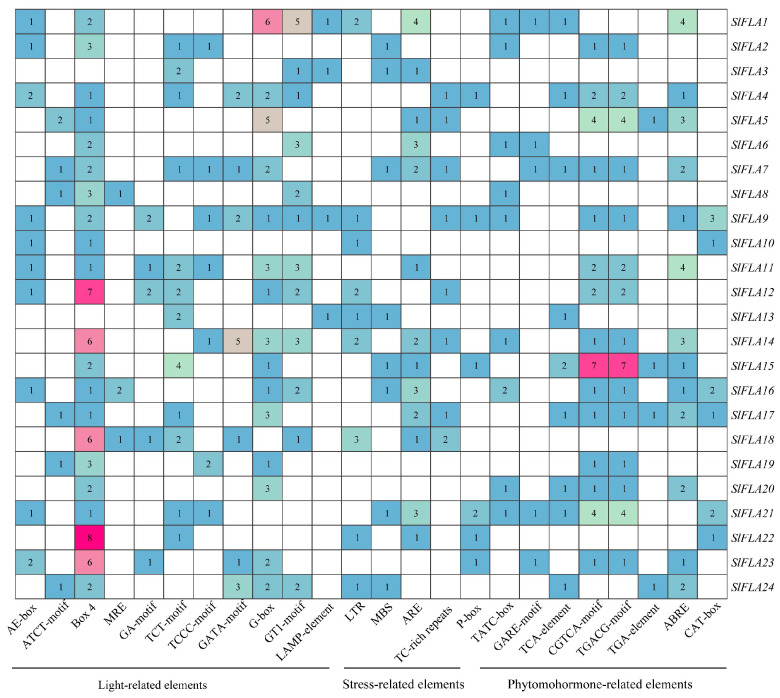
The number of cis-acting elements in tomato *FLA* genes. The different colors and numbers of the grid indicate the numbers of different cis-acting regulatory elements in these *SlFLA* genes.

**Figure 6 ijms-24-16063-f006:**
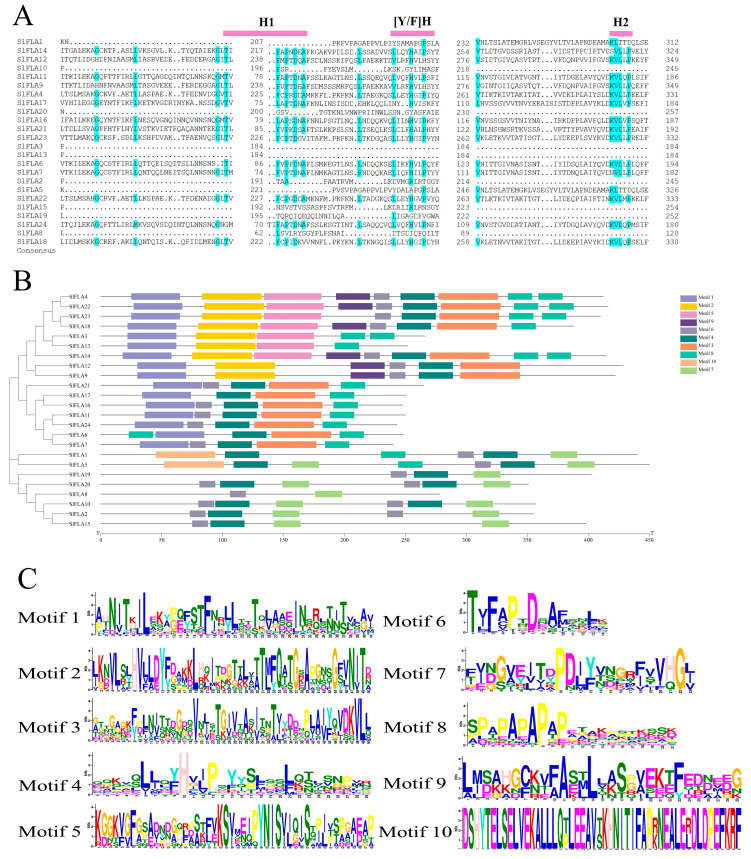
Conserved motif analysis of the *FLA* gene family in tomato. (**A**) Conserved structural domains of the tomato *FLA* gene family. Identical residues are shaded in blue. (**B**) Sequence analysis of *FLA* gene family in tomato. The differently colored rectangles are different motifs. (**C**) Amino acid sequences of different conserved motifs displayed by stacks of letters at each position. The total height of the stack represents the information content of the relative amino acid in the position of each letter in the motif in bits. The height of the individual letter in a stack was calculated by the probability of the letter at that position times the total information content of the stack. The X- and Y-axes represent the width and the bits of each letter, respectively.

**Figure 7 ijms-24-16063-f007:**
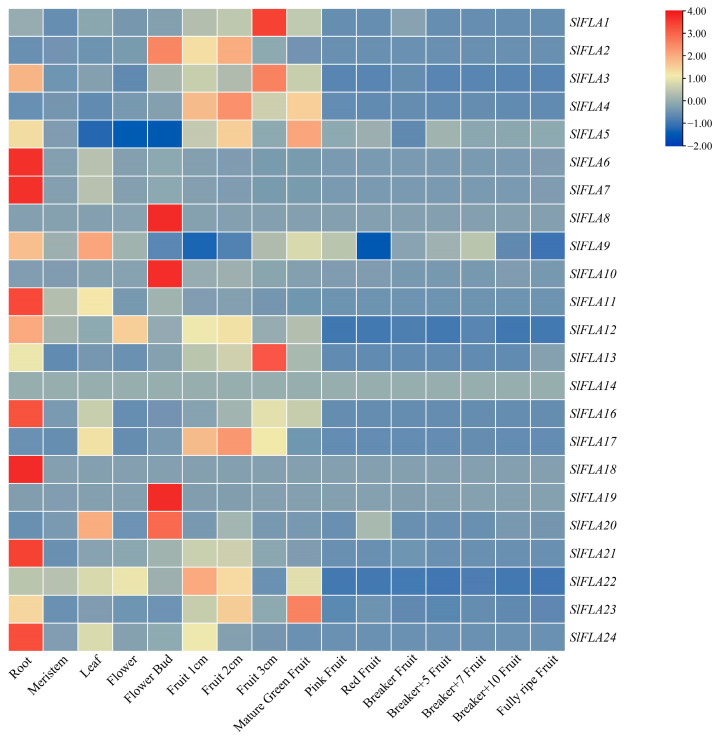
Expression analysis of different tissues of the *FLA* gene family in tomato. Color scale represents fold change normalized by log2-transformed data. Heatmaps are shown in blue/yellow/red for low/medium/high expression, respectively.

**Figure 8 ijms-24-16063-f008:**
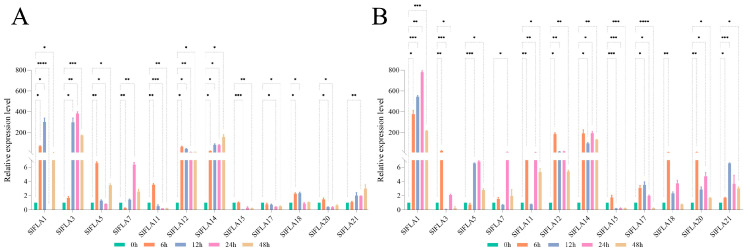
Relative expression analysis of the *SlFLA* gene under the ABA (**A**) and MeJA (**B**) treatments. Error bars represent the standard deviation of the three biological replicates. Different colors indicate different time periods. The asterisk (*) indicates that the expression level of the stress group is significantly different from that of the control group (* *p* ≤ 0.05, ** *p* ≤ 0.01, *** *p* ≤ 0.001, **** *p* ≤ 0.0001, one-way ANOVA, Tukey’s test).

**Figure 9 ijms-24-16063-f009:**
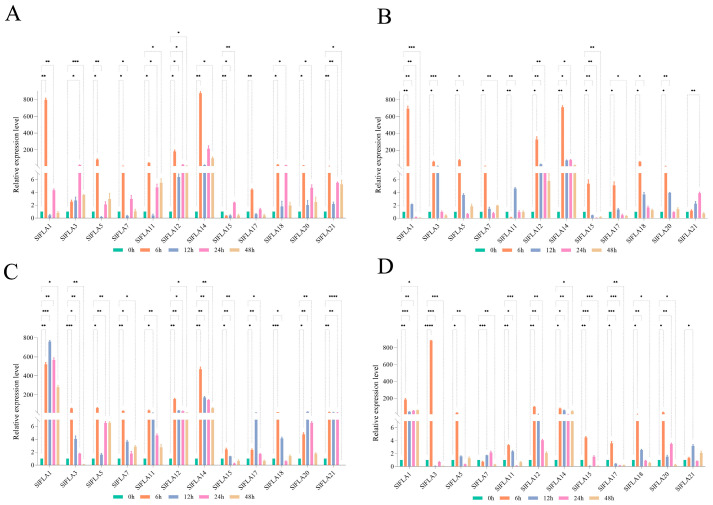
Relative expression analysis of tomato *FLA* genes under the abiotic stresses of cold (**A**), dark (**B**), NaCl (**C**) and PEG (**D**). Error bars represent the standard deviation of three biological replicates. Different colors indicate different time periods. The asterisk (*) indicates that the expression level of the stress group is significantly different from that of the control group (* *p* ≤ 0.05, ** *p* ≤ 0.01, *** *p* ≤ 0.001, **** *p* ≤ 0.0001, one-way ANOVA, Tukey’s test).

**Table 1 ijms-24-16063-t001:** Physicochemical properties of the *FLA* gene family in tomato.

Gene Name	Gene ID	Number of Amino Acid	Molecular Weight	Theoretical pI	Instability Index	Aliphatic Index	Grand Average of Hydropathicity	Subcellular Localization	Signal Peptide	Transmembrane Domain	GPI
*SlFLA1*	101263740	439	48,342.31	5.89	49.1	94.62	−0.215	Chloroplast	Yes	Yes	Yes
*SlFLA2*	101262658	414	43,073.97	5.6	43.06	93.55	0.085	Chloroplast	Yes	Yes	Yes
*SlFLA3*	101262937	427	46,167.18	5.06	46.37	97.47	0.076	Chloroplast	Yes	Yes	Yes
*SlFLA4*	101255812	356	39,469.92	4.54	48.68	94.38	0.114	Extracellular	Yes	Yes	Yes
*SlFLA5*	101244241	249	26,003.19	4.99	36.89	83.09	0.031	Cytoplasm	Yes	Yes	Yes
*SlFLA6*	101249754	421	45,266.73	5.91	52.28	101.16	0.204	Chloroplast	Yes	Yes	Yes
*SlFLA7*	101249482	411	43,611.72	5.61	45.28	89.34	−0.013	Chloroplast	Yes	Yes	Yes
*SlFLA8*	104644532	250	27,513.02	4.89	35.76	93.96	−0.238	Mitochondrion	Yes	Yes	Yes
*SlFLA9*	101249179	350	38,531.73	8.83	52.89	109.74	0.249	Extracellular	Yes	Yes	Yes
*SlFLA10*	101264822	247	26,425.84	6.08	39.01	85.26	−0.29	Chloroplast	Yes	Yes	Yes
*SlFLA11*	101261419	264	28,015.18	6.58	50.2	93.14	0.17	Chloroplast	Yes	Yes	Yes
*SlFLA12*	101252377	409	44,082.41	6.06	31.15	88.7	−0.068	Extracellular	Yes	Yes	Yes
*SlFLA13*	101252705	265	27,776.55	4.83	52.11	83.66	−0.064	Chloroplast	Yes	Yes	Yes
*SlFLA14*	101257010	251	26,706.26	5.89	44.16	82.87	−0.186	Extracellular	Yes	Yes	Yes
*SlFLA15*	104649632	362	36,790.89	4.49	60.78	73.76	−0.096	Endoplasmic ergatoplasm	Yes	Yes	Yes
*SlFLA16*	101256014	247	26,542.2	5.6	46.36	89.6	−0.006	Plasma membrane	Yes	Yes	Yes
*SlFLA17*	101266654	239	26,134.93	8.56	47.59	89.46	0.023	Vacuole	Yes	Yes	Yes
*SlFLA18*	101254871	354	38,682.99	7.16	39.12	101.05	0.289	Vacuole	Yes	Yes	Yes
*SlFLA19*	104644785	449	49,434.64	5.67	46.5	95.75	−0.14	Endoplasmic ergatoplasm	No	Yes	Yes
*SlFLA20*	101244600	415	44,208.4	5.73	38.73	81.9	−0.047	Chloroplast	Yes	Yes	Yes
*SlFLA21*	101249187	397	42,609.93	6.29	49.14	95.52	0.251	Plasma membrane	Yes	Yes	Yes
*SlFLA22*	101265484	401	44,767.54	6.3	44.95	99.68	0.047	Vacuole	Yes	Yes	Yes
*SlFLA23*	101260248	242	25,793.38	5.43	27.98	92.69	0.058	Vacuole	Yes	Yes	Yes
*SlFLA24*	101254060	277	30,716.84	9.34	36.03	83.86	−0.168	Chloroplast	Yes	Yes	Yes

Gene IDs were derived from NCBI’s Gene IDs. The physicochemical properties of the genes were calculated using TBtools. Yes: the sequence has a signal peptide or transmembrane domain or GPI signal. No: the sequence does not have a signal peptide or transmembrane domain or GPI signal.

**Table 2 ijms-24-16063-t002:** Functions of the cis-acting elements of each gene in the tomato *FLA* gene family.

Cis-Element	Number of Genes	Sequence of Cis-Element	Functions of Cis-Elements
ABRE	27	ACGTG	cis-acting element involved in abscisic acid responsiveness
AE-box	12	AGAAACTT	part of a module for light response
ARE	25	AAACCA	cis-acting regulatory element essential for anaerobic induction
ATCT-motif	7	AATCTAATCC	part of a conserved DNA module involved in light responsiveness
Box 4	63	ATTAAT	part of a conserved DNA module involved in light responsiveness
CAT-box	10	GCCACT	cis-acting regulatory element related to meristem expression
CGTCA-motif	30	CGTCA	cis-acting regulatory element involved in MeJA responsiveness
GA-motif	7	ATAGATAA	part of a light-responsive element
GARE-motif	5	TCTGTTG	gibberellin-responsive element
GATA-motif	15	GATAGGG	part of a light-responsive element
G-box	36	CACGTC	cis-acting regulatory element involved in light responsiveness
GT1-motif	26	GGTTAA	light-responsive element
LAMP-element	4	CTTTATCA	part of a light-responsive element
LTR	14	CCGAAA	cis-acting element involved in low-temperature responsiveness
MBS	8	TTTTTACGGTTA	MYB binding site involved in drought inducibility
MRE	4	AACCTAA	MYB binding site involved in light responsiveness
P-box	7	CCTTTTG	gibberellin-responsive element
TATC-box	10	TATCCCA	cis-acting element involved in gibberellin responsiveness
TCA-element	10	CCATCTTTTT	cis-acting element involved in salicylic acid responsiveness
TCCC-motif	8	TCTCCCT	part of a light-responsive element
TC-rich repeats	9	ATTCTCTAAC	cis-acting element involved in defense and stress responsiveness
TCT-motif	20	TCTTAC	part of a light-responsive element
TGACG-motif	30	TGACG	cis-acting regulatory element involved in MeJA responsiveness
TGA-element	4	AACGAC	auxin-responsive element

Number of genes: Total number of such cis-acting elements contained in tomato *FLAs*.

**Table 3 ijms-24-16063-t003:** Ten conserved motif sequences of the tomato FLA protein.

Motif	Width (aa)	Motif Sequence
Motif 1	41	AHNITKILEKYPZFSTFNRLLSTTQLAAEINSRLTITVLAV
Motif 2	50	JKNVLSLHVLLDYFDAKKLHKITDGTTLVTTMFQATGKAPGNSGFVNITD
Motif 3	50	ATNGAGKFPLNITTDGDQVNISTGIVTAKISNTIYDDNPLAIYQVDKVLL
Motif 4	29	ZQKVQLLQYHVJPSYYSLSSLQTLSNPVR
Motif 5	48	KGGKVGFGSADNBGHLPSTFVKSVMEIPYNISVJQISQPJVSPGAEAP
Motif 6	14	TIFAPTDEAFSNLK
Motif 7	23	FINGVEITDPDJYVNGRFVVHGI
Motif 8	21	SPAPAPAPAPETAKAKTKSSK
Motif 9	29	LMSAHGCKVFASLLLASPVEKTFEDBEEG
Motif 10	50	DSHYTELSELVEKALLLQPLEEAVSKHNITIFAPKNEALERDLDPEFKRF

Width (aa): number of amino acids included in the motif. The results were obtained by MEME.

**Table 4 ijms-24-16063-t004:** Secondary structure of the tomato FLA protein.

Protein	Alpha Helix (%)	Extended Strand (%)	Beta Turn (%)	Random Coil (%)	Distribution of Secondary Structure Elements
SlFLA1	35.08	15.26	6.61	43.05	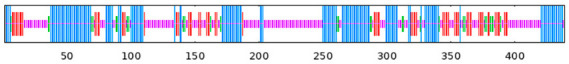
SlFLA2	22.6	25.99	6.5	44.92	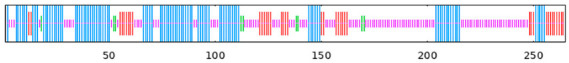
SlFLA3	37.36	14.72	3.4	44.53	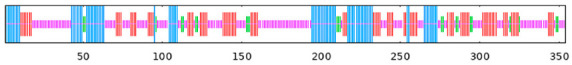
SlFLA4	36.98	16.55	7.06	39.42	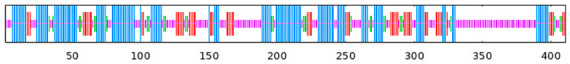
SlFLA5	35.19	18.04	5.12	41.65	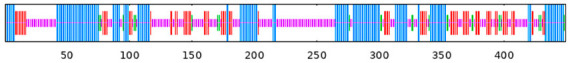
SlFLA6	28.34	19.84	4.86	46.96	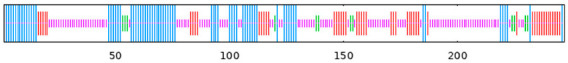
SlFLA7	37.66	16.74	5.44	40.17	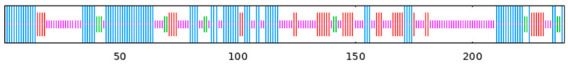
SlFLA8	30.32	22.38	5.05	42.24	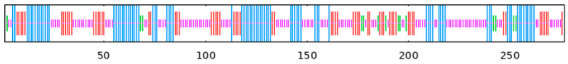
SlFLA9	32.3	19.48	3.8	44.42	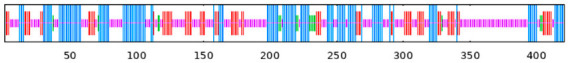
SlFLA10	25	19.94	7.87	47.19	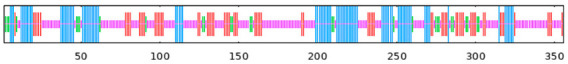
SlFLA11	28.92	15.26	2.41	53.41	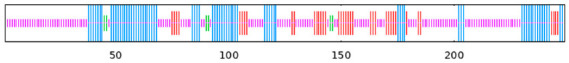
SlFLA12	30.68	19.67	7.26	42.39	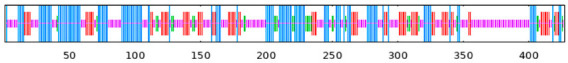
SlFLA13	41.43	13.15	4.78	40.64	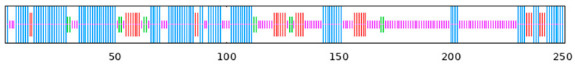
SlFLA14	33.33	18.6	5.31	42.75	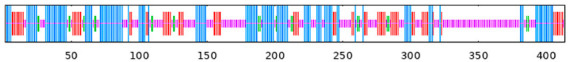
SlFLA15	28.97	15.37	5.54	50.13	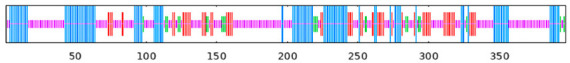
SlFLA16	27.13	18.22	4.05	50.61	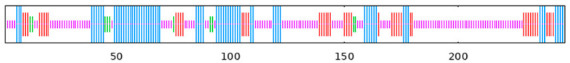
SlFLA17	32.8	20.8	2.8	43.6	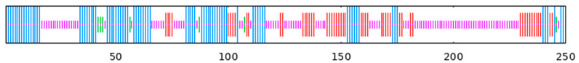
SlFLA18	35.4	19.9	8.01	36.69	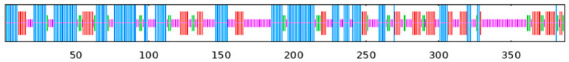
SlFLA19	30.92	21.7	7.23	40.15	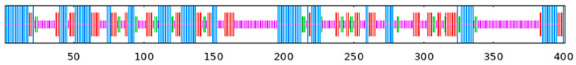
SlFLA20	27.43	18	6.86	47.71	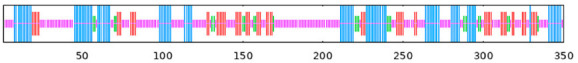
SlFLA21	35.61	17.42	6.06	40.91	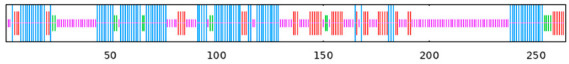
SlFLA22	41.2	14.46	6.27	38.07	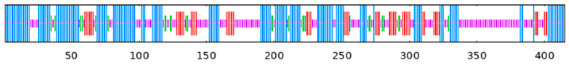
SlFLA23	34.72	19.56	6.36	39.36	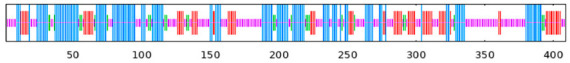
SlFLA24	28.1	23.97	4.96	42.98	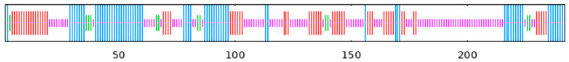

The different secondary structures are expressed as percentages. In the secondary structure figure, blue indicates alpha helix; green indicates beta turn; red indicates extended strand; and pink indicates random coil.

## Data Availability

All data, tables, and figures in this manuscript are original, and are contained within the article and Appendix A.

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
