# Peer review of "Characterization of the FLA Gene Family in Tomato (Solanum lycopersicum L.) and the Expression Analysis of SlFLAs in Response to Hormone and Abiotic Stresses"

_ijms, 2023, doi:10.3390/ijms242216063_

Round 1
Reviewer 1 Report
Comments and Suggestions for Authors
The manuscript entitled „Identification of FLA gene family in tomato (Solanum lycopersicum L.) and functional analysis of SlFLAs in response to hormone and abiotic stresses” by Kangding Yao, Yandong Yao, Zhiqi Ding, Xuejuan Pan, Yongqi Zheng, Yi Huang, Zhuohui Zhang, Ailing Li, Chunlei Wang, Changxia Li, and Weibiao Liao that focuses on fasciclin-like arabinogalacan proteins in tomato. Call wall proteins are important for various processes including response to stresses therefore I found the manuscript timely and valuable. However, several issues need to be solved before the manuscript can be recommended for publication.
1. The title needs to be modified. “Functional analysis” was not performed in this manuscript only gene expression analysis. Functional analysis could be for example analysis of transgenic plants with overexpression of any of SlFLA genes or analysis of knock-out mutants.
2. The overall message that the authors tried to deliver in the manuscript is that the FLA genes have been not described in the genome of tomato. For example, in the abstract, the authors wrote “… it is currently unknown whether FLA exists in tomato…”. This is in fact misleading. In NCBI database search using the term “Solanum lycopersicum + fasciclin-like arabinogalacan” yields 22 hits from the tomato genome. Therefore, I do not think that authors “identified” those genes, maybe the better word is “characterise”. I agree that is valuable to analyse the gene families from plant genomes, especially those containing several genes.
3. More specific keywords should be used.
4. It is not clear what the authors meant by “Therefore, the prediction and study of FLA gene function in horticultural plants could provide important clues to improve plant growth and development.”
5. The aim of the study should be clearly articulated. Moreover, authors should pay attention to using italics for FLA. “FLA” means gene and a gene does not play a role but FLA does (protein).
6. The in silico results could be organised better i.e. first the results about genes (chromosomal localization, intron-exon, promoters) and then the results about proteins.
7. The caption of figures (especially figs. 1, 3, 5, 6, 8) should be more informative. The figure should stand for its own so all details about the figure should be included in the caption. Moreover, the caption of all tables should be also improved. The quality of some figures is low and needs to be improved.
8. Section 2.2 – “For a better understanding of these genes, we analyzed the structure of the proteins encoded by these genes.” – please rephrase
9. Figure 2B – only 10 motifs were found. Or the maximum value was set to 10.
10. Section 2.3. – please explain somewhere what is the basis of the division of FL:A proteins into 5 subgroups.
11. Sections 2.6, 2.7, and 2.8 – the description of the expression analysis results should be more qualitative – what are the differences between the expression of selected variants?
12. The discussion needs to be rewritten. Now the discussion is mostly the repetition of the introduction and results. A more critical assessment of the results is crucial.
13. Page 17, line 402 – what is a high-temperature shaker?
14. Page 17, line 405 – what does it mean “the plants were filled with nutrient solution every 2 d”?
15. Page 17, line 410 – did authors really treat plants with 150 M NaCl?
16. The conclusions should be more conclusive. Maybe authors could propose some physiological roles of some members of the analysed gene family.
Comments on the Quality of English LanguageThe English language needs major improvement. Several sentence is wrong in terms of grammar. Moreover, quite often I could understand what the authors meant.
Author Response
Dear Reviewer,
Thank you for your valuable feedback on our manuscript entitled "Identification of FLA gene family in tomato (Solanum lycopersicum L.) and functional analysis of SlFLAs in response to hormone and abiotic stresses". We appreciate your time and effort in reviewing our work.
We sincerely appreciate your guidance in improving our manuscript. We have revised the manuscript, and would like to submit it for your consideration. According to your comments and suggestions, we have made corresponding changes. In addition, the manuscript has already been carefully checked for language and grammar by https://www.mdpi.com/authors/english. Please find the detailed responses in the attachment file and the corresponding revisions/corrections highlighted in the re-submitted files.
We would like to express our sincere thanks again to you for the constructive and positive comments.
With best wishes,
Yours sincerely,
Kangding Yao, Weibiao Liao

Reviewer 2 Report
Comments and Suggestions for Authors
1. Regarding the figures in the paper that are not very clear, please replace them with high-quality ones.
2. The parts are marked in blue in Figure 2. A should be annotated with their meaning in the illustration section.
3. Regarding the affinities of the FLA proteins, are both represented in Figure 2B and Figure 3. What are the differences between them?
4. Regarding lines 141 to 159, you mentioned the tomato FLA gene. Is the tomato FLA gene unique to tomatoes or is the tomato FLA gene also found in other plants?
5. In line 214, "the expression pattern of SlSLA genes" is incorrect. Please correct it.
6. Please give the full name of "ABA, MeJA, PEG" in the method for better understanding.
7. Regarding Figure 9, FLA gene expression levels were elevated at 6 h and relatively reduced at 12 to 24 h after exposure to low temperature or salt stress. Please add an explanation about this result and add a related discussion in the Discussion section.
8. In the case of fertility regulation and abiotic stresses concerning tomatoes, important and specific genes can be selected for results and it is not necessary to analyze all genes. Data on this section can be placed in the Supplementary file. Please refer to this paper.
Huang, Geng‐Qing, et al. "Characterization of 19 novel cotton FLA genes and their expression profiling in fiber development and response to phytohormones and salt stress." Physiologic Plantarum 134.2 (2008): 348-359.
9. In the paper, you mention “AtFLA11, EgrFLAs” for several times. Does this nomenclature require additional clarification of its meaning in the introduction or discussion? Please refer to this paper and revise it.
10. At line 410, about “ABA (100 μ M)”, please delete the extra space.
Comments on the Quality of English LanguageMinor editing of English language required
Author Response

(The authors gave the same response as above.)

Round 2
Reviewer 2 Report
Comments and Suggestions for Authors
-